# ImpressLearn: Continual Learning via Combined Task Impressions

## Abstract

This work proposes a new method to sequentially train a deep neural network on multiple tasks without suffering catastrophic forgetting, while endowing it with the capability to quickly adapt to unknown tasks. Starting from existing work on network masking (Wortsman et al., 2020), we show that a simple to learn linear combination of a small number of task-specific masks ("*impressions*") on a randomly initialized backbone network is sufficient to both retain accuracy on previously learned tasks, as well as achieve high accuracy on new tasks. In contrast to previous methods, we do not require to generate dedicated masks or contexts for each new task, instead leveraging transfer learning to keep per-task parameter overhead negligible. Our work illustrates the power of linearly combining individual *impressions*, each of which fares poorly in isolation, to achieve performance comparable to a dedicated mask. Moreover, even repeated *impressions* from the same task (homogeneous masks), when combined can approach the performance of heterogeneous combinations if sufficiently many *impressions* are used. Our approach scales more efficiently than existing methods, requiring orders of magnitude fewer parameters and can function without modification even when task identity is missing. In addition, in the setting where task labels are not given at inference, our algorithm gives an often favorable alternative to the entropy based task-inference methods proposed in (Wortsman et al., 2020). We evaluate our method on a number of well known image classification data sets and architectures.

**Keywords:** Catastrophic forgetting, Continual learning, Neural networks, Masking

## 1 Introduction

Sequential learning without catastrophic forgetting has been an area of active research in machine learning for some time (Maes et al., 1996; Thrun & Pratt, 1998; Serra et al., 2018). A precondition for achieving artificial general intelligence (AGI) is that models be able to learn and remember a wide variety of tasks sequentially, without forgetting previously learned ones. In real-world scenarios, data from different tasks may not be available simultaneously, which makes it imperative to both allow continued learning as well as to avoid catastrophic forgetting of a potentially unbounded number of tasks (see also *The Sequential Learning Problem* (McCloskey & Cohen, 1989), *Constraints Imposed by Learning and Forgetting Functions* (Ratcliff, 1990) and *Lifelong Learning Algorithms* (Thrun & Pratt, 1998)). Recently, some successful approaches to combat this problem use task specific submodels, which allow neural networks to context-switch between different learning tasks (Wortsman et al., 2020; Mallya et al., 2018; Mancini et al., 2018). The underlying context for each task can be represented as "*filters*" or "*masks*", altering the network connections for each task. Yet all of these approaches scale unfavorably with the number of unique tasks to be learned.

**Our contribution:** We propose a novel method which exploits transfer learning and network masking to sequentially learn a theoretically *unlimited* number of tasks with much lower parameter overhead than prevailing benchmarks. Our method, termed *ImpressLearn*, uses elements from *Supermasks in Superposition (SupSup)* by Wortsman et al. (2020). The *SupSup* approach leverages the observation that even within randomly weighted neural networks there exist task-specific supermasks—subnetworks produced by overlaying a binary mask that selectively removes connections—which achieve good performance on the task. These supermasks can be learned from task-specific data and stored, one

mask per task. At inference, the appropriate task-specific mask is applied when task identity is known. When task-labels of a previously seen task are not known, the correct mask can be inferred via entropy considerations.

**A "basis" of masks:** Our approach goes further, driven by two core ideas. In order to dramatically improve *scaling* of the parameters with the number of different tasks, we reduce the number of necessary masks to a constant number, independent of the number of different tasks. Our *basis-masks* are constructed from a small number of initial tasks. Leveraging *transfer learning*, this set of learned masks, each of which can be interpreted as an "*impression*" of a previously seen prior task, serves as a set of latent features for new learning objectives, encoding common structural information. Our second idea is to use the power of *linear combinations* to combine these *impressions* to quickly learn a real-valued mask that performs well for *new* tasks. Hence, other than a fixed number of basis-masks, only a small number of coefficients need to be learned and stored for each subsequent task. This greatly benefits scalability, a major drawback of previous approaches, as this in principle allows for an unlimited number of new tasks while the parameter overhead is several orders of magnitude smaller than storing even a compressed binary network mask.

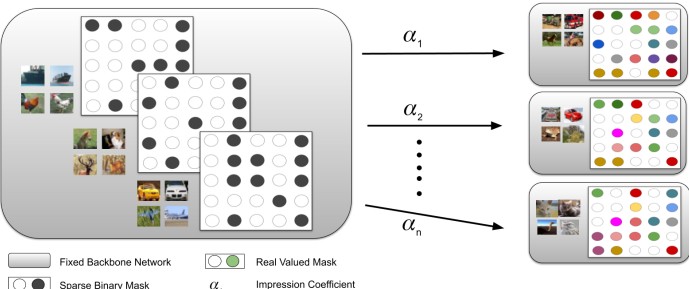

Figure 1: Overview of the ImpressLearn method (best viewed in colour). Given a fixed-weight randomly initialized backbone network, we show that a few binary mask "impressions" from training prior tasks (3 in this case) can form a linear combination with coefficients $\alpha$ to learn new tasks. Unlike the impressions, the resulting mask is real-valued, and not binary.

Specifically, *ImpressLearn* uses the first few tasks from a multi-task set to generate a relatively small number of binary *masks*. For subsequent tasks, these impressions are used akin to a *basis*: new tasks are learned through a simple linear combination of the basis-impressions. This might be reminiscent of associative learning where impressions of previous scenarios are combined to cope with new ones. In order to increase the expressivity of this approach, we allocate one real-valued coefficient for each *layer of weights* of the underlying model. Learning these linear combinations is fast and simple; and storing the coefficients requires only trivial overhead per task. For instance, in the case of LeNet-300-100 on `RotatedMNIST` with a basis set of 10 masks, new tasks require only $3 \times 10 = 30$ parameters while a new SupSup mask at e.g. $10\%$ sparsity requires $\geq 25K$ weight indices to specify. We provide ample empirical evidence of the efficacy of our *ImpressLearn* approach on a variety of benchmarks, outlining the radical savings in parameters needed to be stored per task compared to SupSup.

**Homogeneous masks:** Somewhat surprisingly, we can even generate all basis-masks from the *same* initial task using different random seeds for the learning algorithm (but the same random backbone network). We show that with a sufficiently large number of such *"homogeneous"* impressions, our algorithm learns linear combinations with close to benchmark accuracy on new tasks. We are reminded of an infant learning by taking different "snapshots" of the same object, to infer properties of another. This "homogeneous impression" method is particularly useful to address possible drifts in the data. Akin to *ensembling*, it leverages the power of linear combinations for transfer learning. An additional important advantage is that the homogeneous method has no limit on the number of basis-mask we can generate ab initio (since this requires a single task only).

To provide another benchmark for our approach, we also try our linear combination approach with a set of completely *random* masks of desired sparsity. We demonstrate on several benchmarks that if we chose a sufficiently large set of such random basis masks, our optimization still yields good accuracy. While naturally, combinations of random masks lag behind the heterogeneous and also

the homogeneous setting, we show that there is a trade-off between number of masks and their task-specificity (non-randomness). In settings where producing task-specific basis masks is costly and where we might not wish to perform the SupSup mask search, our linear combination optimization still can yield satisfactory results with a large number of random masks.

**Example: LeNet-300-100 on RotMNIST** To illustrate our approach and its performance, Fig. 2 shows the accuracy of our *ImpressLearn* approach, compared to *SupSup* on the `Rotated MNIST` task set. This allows us to share some observations which generalize to a wide variety of benchmarks (see Sec. 4). First, to show the power of the linear combination approach beyond pure transfer learning, we evaluate the accuracy of a basis mask on tasks different from the one the mask was trained on: we get close to random accuracy ("X" in Fig. 2, leftmost box plot). Next we illustrate how a linear combination of a small number of masks from different tasks (heterogeneous impressions) achieves close to benchmark accuracy on new tasks. In this case, combining a few masks already gives performance close to, even exceeding, the SupSup benchmark. This is followed by the performance of a set of *homogeneous* masks, where all impressions are generated from the first task (middle of Fig. 2). We can see that when using homogeneous impressions, we need a larger number of basis masks to achieve accuracy comparable to the heterogeneous setting, but that ultimately here also we match SupSup accuracy, with a vastly smaller number of parameters than the SupSup approach. Lastly, in the right part of Fig. 2 we study how our linear combination approach performs on tasks from the basis set when task labels are not provided at inference (*basis*-**GN** setting). Here, the our optimization finds the "correct" mask, or a linear combination of equal or better accuracy, providing an alternative to the entropy-based **GN**-inference of (Wortsman et al., 2020).

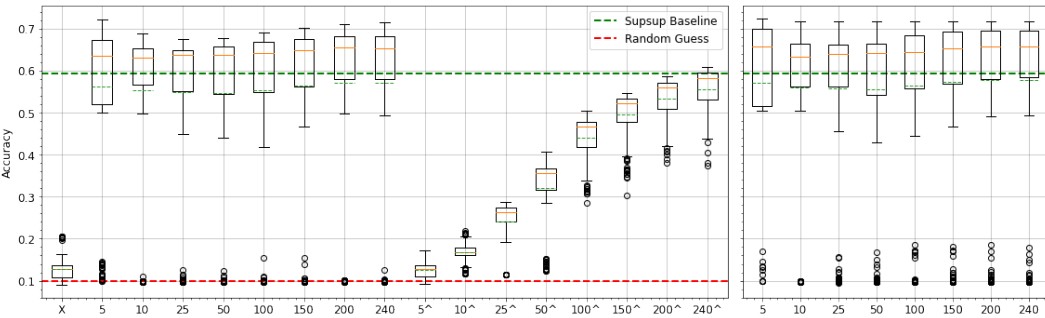

Figure 2: Left: Average validation performance on 10 *new* Rotated MNIST tasks by number of impressions and impression type. $X$ indicates incorrect mask. Numbers $5, 10, \ldots$ indicate *heterogeneous* and numbers with $^\wedge$ indicate a *homogeneous* impression set. Right: Average performance on tasks 1 to $\min(|\mathcal{M}_{hom}|, 25)$ in *basis*-**GN** regime on basis tasks without task identity. All results are averaged over 3 different seeds and masks of density $\in \{5, 8, 10, 15, 20, 25, 30\} \times 10^{-2}$.

**Unknown task labels at inference and a new GN algorithm:** Wortsman et al. (2020) showed that in the so called **GN** regime where task labels are given at training but not at inference, entropy minimization starting from a uniform combination of (binary) masks quickly singles out the "correct" mask for a task. Even though our masks for non-basis tasks are real-valued linear combinations of binary masks, we show that in the **GN** setting a similar entropy minimization allows us to infer the correct real-valued mask with the same approach. Moreover, in what we call the *basis*-**GN** regime where the unlabeled tasks come from the set of basis tasks, we show that our *ImpressLearn* optimization finds a linear combination that either favors the correct mask or yields an even better accuracy than the corresponding basis mask without using the entropy of the output. This means that the optimization routine of *ImpressLearn* gives an alternative algorithm for inference in the case of unknown task labels at test time: it finds a linear combination of masks that often outperforms the corresponding basis mask.

In Section 2, we briefly review related work and general approaches to countering catastrophic forgetting, highlighting research that motivated our approach. In Section 3, we lay out the *ImpressLearn* algorithm, including our modified objective function. In Section 4, we demonstrate the effectiveness of *ImpressLearn* on a variety of benchmark data sets and architectures to show close-to-benchmark performance with a drastically reduced parameter count on new tasks, especially where the number of possible tasks is large. We discuss our results including the trade offs between homogeneous and

heterogeneous masks. Finally, in Section 5 we talk about limitations of our work, and areas for future research.

## 2 RELATED WORK

In practice, intelligence systems should be able to learn a variety of tasks incrementally and without experiencing *catastrophic forgetting*—degrading performance on previously acquired skills (McCloskey & Cohen, 1989), while possibly transferring current knowledge to facilitate future training. Oftentimes tasks are diverse and not available concurrently, making joint training impractical. Conversely, ordinary finetuning (continued training of a pre-trained network) inevitably leads to catastrophic forgetting. *Continual learning* encompasses a broad spectrum of algorithms and architectures that address these issues and propose systems capable of learning from an incremental stream of tasks while minimizing catastrophic forgetting. Most naturally, these techniques are categorized into three groups described below (Delange et al., 2021; Wortsman et al., 2020).

**Regularization-based methods:**   This class of algorithms trains on new tasks by finetuning weights but attempts to retain performance on previously learned tasks through regularization. A number of studies assess the importance of individual parameters for previous tasks and penalize their displacement accordingly during optimization. Pioneering this approach, Serra et al. (2018) estimates parameter importance using a Laplace-approximated posterior distribution after training on earlier tasks. Zenke et al. (2017) impose a quadratic penalty proportional to the accumulated sensitivity of previous loss functions to perturbations in the corresponding parameter; Aljundi et al. (2018) use the same strategy but accumulate sensitivity of the network output to parameter perturbations instead. In contrast, Li & Hoiem (2018) regularize by means of distilling current knowledge on the incoming task's data and using it during finetuning. Regularization-based methods require no additional memory overhead per task and hence are advantageous in capacity constrained settings. However, the plasticity of a network decreases with more tasks, imposing a natural limit on new tasks, which creates a trade-off between learning new tasks and catastrophic forgetting.

**Replay methods:**   Techniques in this category preserve performance on prior tasks by replaying, rehearsing or otherwise utilizing representative samples from the corresponding data distributions. Most commonly, replay models store examples of seen data in a separate memory buffer (Rebuffi et al., 2017; Rolnick et al., 2019; Riemer et al., 2019); others maintain generators that approximate the original data distribution and provide pseudo-examples (Atkinson et al., 2021; Shin et al., 2017). While the majority of algorithms in this group replay stored examples during optimization to mimic joint training, Lopez-Paz & Ranzato (2017) uses them to constrain optimization space and ensure positive knowledge transfer. Replay methods require additional memory to store data samples or allocate generators, however, these costs are usually kept fixed. For this reason, like regularization-based models, replay models exhibit poor stability-plasticity trade-off with more tasks, but often come with increased memory requirements when compared to regularization based methods.

**Parameter isolation methods:**   These methods allocate new parameters for incoming tasks and feature little to no interference between previously learned tasks. Rusu et al. (2016) allocate a new copy of the network and enable forward transfer learning with lateral connections going into new modules. Ren et al. (2017) combine individual learners in a decision tree and eliminate outdated models with tree pruning. A large body of recent algorithms piggyback on a single backbone network shared by all tasks. As such, Wen et al. (2020) (*BatchEnsemble*) operate on a fixed pretrained network and, for each incoming task, optimize for a rank one parameter mask applied to the backbone at inference. Mallya & Lazebnik (2018) (*PackNet*) use pruning to assign subsets of free parameters of a backbone network to individual tasks by issuing one binary parameter mask per task. Assigned parameters are forever frozen at their trained values, limiting capacity of the network for future tasks. In a subsequent study, Mallya et al. (2018) (*Piggyback*) lift this limitation by directly optimizing per-task binary masks and applying them to a fixed pretrained network.

**SupSup**   Our *ImpressLearn* algorithm is most closely related to yet another similar method called *SupSup* (Wortsman et al., 2020). This algorithm trains individual per-task binary masks but applies them to a randomly-initialized network, leveraging the existence of supermasks (Zhou et al., 2019). The mask optimization algorithm, *edge-popup* (Ramanujan et al., 2019), uses a heaviside function to

binarize mask values on the forward pass and employs a straight-through estimator when computing gradients. In addition, Wortsman et al. (2020) propose different training and inference modes depending on availability of task identifiers; e.g., **GG** refers to the scenario when task identifies are known during both training and inference, while in **GN** they are available only during training. For the **GN** case Wortsman et al. (2020) introduce a *one-shot* algorithm to infer task identity by minimizing the output entropy, starting from a uniform linear combination of masks and optimizing the coefficients. While at first glance this algorithm resembles ours, there are essential differences as we use optimization of a refined linear combination to learn *new* tasks. While SupSup and other related methods suffer no catastrophic forgetting regardless of the number of tasks, they are required to store the corresponding parameter mask for each task, which is costly. SupSup try to address this by storing masks as attractors of a Hopfield network, but it is unclear how feasible this approach is.

## 3 APPROACH

**Preliminaries and notation.** We largely adopt the notation from Wortsman et al. (2020). For the standard $l$-way classification task from a set of tasks $T$, inputs $x$ are mapped to a distribution $p$ over output neurons $\{1, ..., l\}$. Let $f$ be a network architecture defined over the backbone weight matrix $W$, which is taken to be random but fixed. Similar to (Wortsman et al., 2020) we use the Edge-Popup training algorithm of Ramanujan et al. (2019) (based on earlier work by Zhou et al. (2019)) to train a binary mask $M^t$ for a task $t \in T$. We further stratify each mask by layer: let $d$ be the number of weight-layers of the network computing $f$. For $i \in [1, \ldots, d]$ denote by $M_i^t$ the part of the binary matrix corresponding to layer $i$ of the network such that $M^t = \oplus_i M_i^t$. Similarly, let $W_i$ denote the submatrix of $W$ corresponding to the $i$th layer. The *sparsity* $\in (0, 1]$ of a mask $M^t$ is given by the fraction of 1s in the mask and is usually fixed in advance to $s$[1]. For each task $t \in T$ the SupSup algorithm aims to find a mask $M^t$ of sparsity $s$ to minimize $E_{(x,y)} \mathcal{L}(y, f(x, M^t \odot W))$, where $\mathcal{L}$ is the model's loss function (cross-entropy in most cases), $y \in \{0, 1\}^l$ is the one-hot encoding of the label and $\odot$ defines the element-wise product.

*Supermask training with edge-popup:* For each task $t$, the edge-popup algorithm in SupSup learns a score matrix $S$ with the same dimensions as $W$ via gradient descent as a function of the (unchanged) weights and the loss function. It then generates the mask $M$ by setting the top $s$ fraction of scores in $S$ to 1 and the rest to 0.

*Inference without task labels (GN):* The task inference idea presented in Wortsman et al. (2020) is based on the intuition that the correct mask for a task should give a highly certain (i.e. low entropy) model output. The algorithm hence starts with an equally weighted combination of masks and uses gradient descent on these coefficients using output entropy as the loss function. This quickly singles out the correct mask by up-weigting the corresponding coefficient. We adopt this algorithm without change to our real-valued masks in the **GN** scenario.

**The ImpressLearn algorithm.** For *ImpressLearn* we define a set *basis* tasks $T_b \subset T$ and call its compliment $T_n = T \setminus T_b$ the set of "new" tasks. We randomly initialize and freeze once and for all a *backbone* network with weights $W$[2].

Step 1: For each $t \in T_b$ we use the edge-popup algorithm as in SupSup to create one or several basis-masks $M^t$, leading to a set $\mathcal{M} = \{M^1, M^2, \ldots\}$ of *basis-masks*. We use up to 250 basis masks for various benchmark architectures and data sets.

Step 2: For each new task $s \in T_n$ we define a *coefficient-matrix* $\alpha^s \in \mathbb{R}^{|\mathcal{M}| \times d}$. To find the layerwise linear combination of basis masks, we optimize

$$\hat{\alpha}^s = \arg\min_{\alpha^s} \mathcal{L}\left(y, f\left(x, \sum_{M^t \in \mathcal{M}} \oplus_{i=1}^d \alpha_{t,i}^s (M_i^t \odot W_i)\right)\right). \tag{1}$$

---

[1]Note that often sparsity is defined in a complementary way as the fraction of zeros in $M^t$, but we will keep consistency with Wortsman et al. (2020). We consider sparsities $\in [0.05, 0.5]$, where 0.05 means 95% of weights are deactivated.

[2]Various standard initializations are possible; we use a Kaiming normal distribution (He et al., 2019) and set the biases to zero.

**Initial Condition on the** $\alpha_{t,i}$**:** A priori all basis masks have equal chance to contribute to a new task. Moreover, testing a basis mask for task $t$ on any other task $t' \neq t$ gives close to random performance, meaning there is no direct knowledge transfer (see Section 4). Hence, a *uniform prior* on the distibution of the $\alpha$-coefficients per layer is a reasonable assumption. We treat each layer independently, setting $\alpha_{i,t} = 1/|\mathcal{M}|$ at the start of the gradient-based optimization.

**Regularization:** While overfitting is less of a concern, given the small number of parameters in our optimization Eq. (1) we want to enforce that our algorithm aims for a *sparse* combination of masks when possible, especially when faced with one of the basis tasks, where we expect our algorithm to identify the "correct" mask among the basis masks. For some benchmarks we hence apply an L1-penalty for each layer for deviations from *unit* L1-coefficient norm, to obtain the loss function:

$$ J = \mathcal{L} + \lambda \sum_{i=1}^{d} \left( \sum_{t=1}^{|\mathcal{M}|} |\alpha_{t,i}^{s}| - 1 \right)^2 \tag{2} $$

**Heterogeneous vs homogeneous masks, random masks and overlap:** In our *heterogenous* approach, we create one mask for each task in $T_b$. In this setting we leverage knowledge transfer from *all* basis tasks for a new task. However, in some settings with few tasks, limiting mask generation to one per task affects the viability of our method and doesn't allow for scaling benefits to become apparent. For example, for Split-CIFAR-100 with 20 taks, we can only generate at most 20 heterogeneous masks. We have hence also evaluated our approach with a set of basis masks all coming from the same task, so called *homogeneous* masks. A priori it is unclear whether masks produced on the same backbone network for the same task are sufficiently different to generate a diverse enough basis set. While previous work suggests that wider network architectures can support multiple suitable subnetworks for a given task Ramanujan et al. (2019); Frankle & Carbin (2019), we find this effect is prominent even using relatively conservative architectures such as LeNet 300-100 with Permuted MNIST (Lecun et al. (1998)). Our experiments show that masks are very sensitive to initial conditions of the popup scores and to data ordering: the overlap of homogeneous masks produced with different random seeds is close to the overlap of randomly picked masks. Hence, at sufficiently low sparsities, homogeneous masks, even on the same backbone network, are close to independent.

Our approach can be extended to a mix of homogeneous and heterogeneous impressions, though in this paper we only study the trade-off for *"pure"* homogeneous and heterogeneous settings.

To quantify the importance of task specific data in mask generation we also compare *ImpressLearn* with the scenario where in Step 1 the basis masks are picked *randomly* instead of through the SupSup popup algorithm (see Sec. 4). This allows us to study the trade-offs between random and task-specific tasks and gives an alternative in the case where the SupSup mask search is hard to implement. In this setting, for a fixed sparsity, masks are chosen uniformly at random (by picking edges one at a time without repetition until the desired sparsity is reached). This mask set is then used for Step 2 of *ImpressLearn* to optimize the $\alpha$-coefficients. While in general this approach requires a larger base set than SupSup-based approaches, it avoids the SupSup optimization step, which in some scenarios could be costly (since it requires optimization of a larger number of parameters than our Step 2).

**Optimization and parallelization:** We use the SupSup algorithm with random initializations for the edge popup scores and data ordering to generate the basis mask set $\mathcal{M}$. One fetching attribute of using a set of basis masks is the ability to *parallelize* the training process. Our approach allows for all masks to share the same backbone network, both in the heterogeneous and the homogeneous setting. Thus one can parallelize the same model and train multiple tasks at the same time on different GPUs, only constrained by the number of cores or GPU accelerators available.

To optimize the $\alpha^s$ for *new* tasks, we use stochastic gradient descent.

## 4 EXPERIMENTAL RESULTS

### 4.1 DATA, MODELS AND PARAMETERS

We evaluated *ImpressLearn* over the following classification datasets: MNIST (Lecun et al., 1998): Permuted and Rotated; Split CIFAR-100 (Krizhevsky, 2012) and Split ImageNET (Deng et al., 2009). A detailed description of our experimental choices and infrastructure can be found in App. A.1.

## 4.2 RESULTS

Our results are presented in Figs. 2 (Sec. 1), 4, 3 and 5. We have also performed a couple of sanity checks to evaluate our algorithm and understand which ingredients contribute to its performance.

**Evaluating with "incorrect" masks** To make sure that any accuracy improvements from the linear combination are an effect of the optimization routine of the $\alpha$ coefficients itself, and do not come from chance application of a suitable mask that does well on other tasks, we test impressions derived from one task on other tasks. We find that the incorrect mask fails to have any predictive power on other tasks when used in isolation, giving a roughly random accuracy of $10 \pm 3\%$ for Permuted/Rotated-MNIST/SplitImageNet (Figs. 3, 2 and 5) and $20 \pm 2\%$ on SplitCIFAR (Fig. 4) benchmarks respectively.

**Performance on unlabeled basis tasks - *basis*-GN setting:** To evaluate the strength of our optimization routine, we have studied the "basis"-**GN** setting, by which we mean that our model is presented with one of the *basis* tasks without explicitly providing a task label (as opposed to the full **GN** setting where the unlabeled tasks can come from the basis and the new set). We show that for most benchmark our algorithm is able to either find the correct mask or a linear combination of basis masks of equal or often better accuracy. As such, our $\alpha$-optimization algorithm can serve as an *alternative* to the one-shot entropy based algorithm of SupSup for task inference in the GN setting. More details and a comparison for PermutedMNIST are given in App. A.3.

**Results:** Overall, our results demonstrate the strength of *ImpressLearn* in various settings. We see that the *ImpressLearn* coefficient optimization is able to learn new tasks well with a fraction of the parameters required by other approaches. In line with expectations, we see a strong positive relationship between accuracy and the size of the impression set $|\mathcal{M}|$, with saturation reached at different data-specific sizes. While the number of heterogeneous masks required for good accuracy varies by dataset, *ImpressLearn* is particularly resource efficient when the number of possible tasks is *large* and it becomes costly to store a separate mask for each task (e.g. RotatedMNIST and PermutedMNIST). We also see that compared to the heterogeneous scenario we need more homogeneous basis masks to achieve similar performance (and even more random masks, see App. A.4).

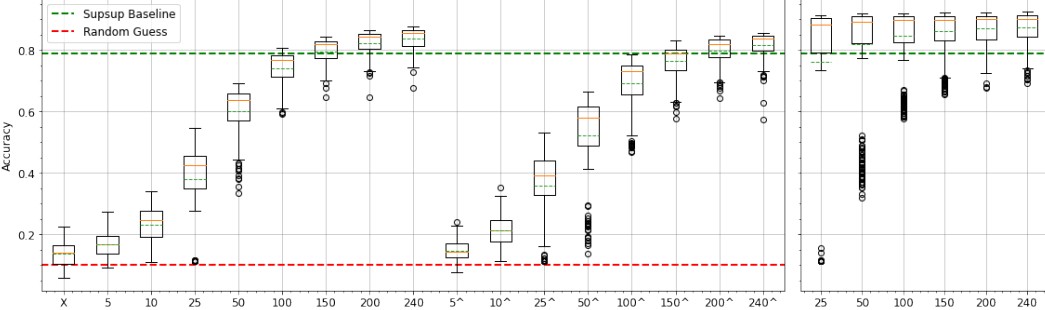

Figure 3: Left: Average performance on 10 new `PermutedMNIST` tasks by number of impressions and impression type. $X$ indicates incorrect mask. Numbers $N$ indicate a heterogeneous and $N^\wedge$ a homogeneous impression set. Right: Performance of our $\alpha$-optimization in the *basis*-**GN** regime averaged over $min(|\mathcal{M}|, 25)$ basis tasks. All results are averaged over 3 different seeds and masks of density $\in \{5, 8, 10, 15, 20, 25, 30\} \times 10^{-2}$.

For the SplitImageNet task we only show results for a heterogeneous basis mask set of $\leq 35$ masks due to limitations on compute. We anticipate that a larger basis mask set will achieve performance closer to the SupSup benchmark and highlight again that our approach scales to an *unlimited* number of tasks with few extra parameters. Note also that for SplitImageNet, in the *basis*-**GN** regime our $\alpha$-optimization yields a better accuracy than the SupSup baseline or the entropy based approach of (Wortsman et al., 2020), highlighting the power of our linear combination routine.

**GN setting:** We have verified on several benchmarks that the one-shot algorithm of Wortsman et al. (2020) continues to work in our setting where the masks for new tasks can be viewed as *real-valued* masks obtained through linear combinations of binary (basis) masks. Our experiments show that the

one-shot algorithm continues to work without change. For PermutedMNIST for example, in the **GN** scenario the algorithm was able to find the correct mask 94.6 % of the time in 3 epochs, and 98.1 % within 5 when inferring identity among 250 learned tasks.

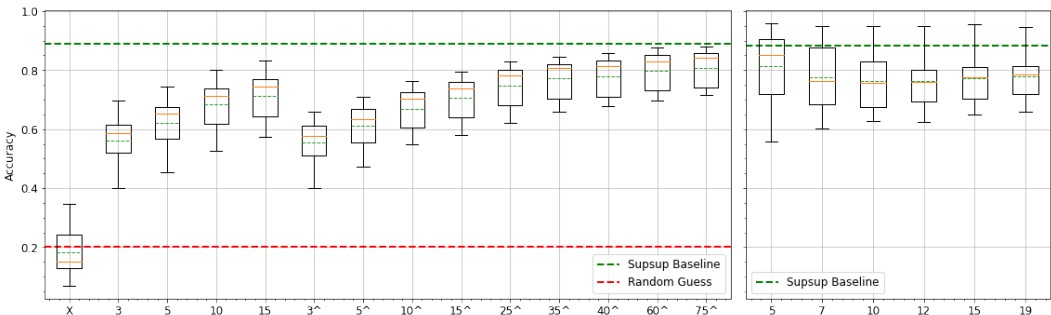

Figure 4: Left: Average performance on 5 new Split CIFAR-100 tasks by number of impressions and impression type. $X$ indicates incorrect mask. Numbers $N$ indicate a heterogeneous and $N^\wedge$ a homogeneous impression set. Right: Performance of our $\alpha$-optimization in the *basis*-**GN** regime averaged over $min(|\mathcal{M}|, 25)$ basis tasks. All results are averaged over 3 different seeds and masks of density $\in \{5, 8, 10, 15, 20, 25, 30\} \times 10^{-2}$.

Table 1: Parameter count for different architectures and data sets.

| Dataset (Max Tasks)
Model | **O**(params) | Mask (kB) | $\|\mathcal{M}\|$ | Layers | $\|\alpha^t\|$ | $\Phi$ (kB) | $\mathcal{C}$ x |
|---|---|---|---|---|---|---|---|
| PermutedMNIST (784 !) | | | | | | | |
| LeNet 300-100 | 266K | 64.9 | 100 | 3 | 300 | $\approx 1.17$ | **55.5** |
| RotatedMNIST (359) | | | | | | | |
| LeNet 300-100 | 266K | 64.9 | 5 | 3 | 15 | 0.96 | **67.4** |
| Split CIFAR-100 (20) | | | | | | | |
| ResNet-18 | 6.2M | 1513 | 10 | 21 | 210 | 757.6 | **1.99** |
| Wide ResNet-18[†] | 11.7M | 2856 | 10 | 21 | 210 | 1429 | **1.99** |
| Wide ResNet-34[†] | 21.8M | 5322 | 10 | 37 | 370 | 2662 | **1.99** |
| ImageNET (2100) | | | | | | | |
| ResNet-50 | 25.6M | 6250 | 75 | 53 | 3975 | 238.7 | **26.2** |
| ResNet-101[†] | 44.5M | 10864 | 75 | 104 | 7800 | 418.47 | **25.9** |
| ResNet-152[†] | 60.2M | 14697 | 75 | 155 | 11625 | 570.31 | **25.7** |
| pre-ResNet-200[†] | 64.7M | 15795 | 100 | 203 | 15225 | 623.6 | **25.3** |
| VGG 16[†] | 137.9M | 33667 | 75 | 16 | 1200 | 1207 | **27.89** |
| WRN-50-2-bottleneck[†] | 68.9M | 16821 | 75 | 53 | 3975 | 616.28 | **27.2** |

Table 2: Parameter tradeoff for different models and datasets. The maximum number of possible tasks for each data set is given in brackets in the first column. For ImageNet we have assumed that entire set of classes is split into 10-way tasks. For RotatedMNIST we have assumed a 1 degree granularity of rotations. **O**(params) is the number of parameters of the network (weights only, assuming biases are set to zero). Mask Size is the space on disk required to store each additional SupSup mask as 16-bit integers. $|\mathcal{M}|$ was approximately chosen to give either benchmark or very good performance on new tasks. $|\alpha^t|$ is the number of parameters (floating point) per additional new task, given a basis of size $|\mathcal{M}|$. $\Phi$ is the storage per task amortizing the cost of storing $|\mathcal{M}|$ basis masks over all possible tasks. $\mathcal{C}x$ is the compression or savings ratio compared to SupSup. † marks those architectures we have not run due to resource constraints. For those we have interpolated the size of the basis set $|\mathcal{M}|$ from those architectures/data sets that we ran.

### 4.3 MODEL EFFICIENCY/PARAMETER SAVINGS

Here, we present the parameter savings for our *ImpressLearn* approach, compared to the SupSup proposal for various architectures and data sets. Our approach has the fixed cost of storing the basis masks as well as the task-specific $\alpha$ coefficients for each new task. We amortize the basis masks across all tasks for comparison.

Table 2 highlights the parameter savings of our approach for various data sets and architectures. Amortizing mask storage across tasks and accounting for storage of the $\alpha$ coefficients, we get particularly impressive savings of an order of magnitude or more for data sets with a large number of tasks, like PermutedMNIST and (general) ImageNet. We have also evaluated potential savings our approach would yield for several common architectures by giving an educated guess on the number of required basis masks. In the case of ImageNet the number of potential tasks is so large that storage of basis masks per new task is very small even if we double the number of masks considered, and memory required to store the $\alpha$ remains small. Overall, we believe our approach affords considerable savings in memory at the expense of either no or small loss in accuracy.

## 5 DISCUSSION

In this work, we propose a novel way to adapt an existing continual learning algorithm, leveraging principles from transfer learning to generalize to new tasks, allowing for scalable and parameter-efficient continual learning. Using a simple linear combination of masks, or impressions, we see that even this basic setup is able to learn new tasks effectively. We show that this effect is consistent across task types and network architectures, and that it achieves competitive performance while using significantly fewer parameters. This work highlights the advantages of re-using existing meta-features learned on previous tasks for future learning problems and opens up a space of possibilities of applying transfer learning to protect against catastrophic forgetting.

One application of our approach could be protection against *drifts* in the data. In the homogeneous setting, in anticipation of drift, one could learn a few basis masks and a linear combination, and then update the latter accordingly when anticipating that the underlying data changes slightly but steadily. To our knowledge, no other approach allows for such easy continued adjustments, as most of them fix masks or weights for fixed tasks.

We have tried several variations of our approach using different *types* or masks:

*Signed Binary Masks*: $M^t \in \{-1, 0, 1\}^{|M^t|}$. Allows for a little more flexibility but did not perform as well as 0,1 masks.

*Real valued masks*: $M^t \in \mathbf{R}^{|M^t|}$. Real valued masks performed much better at high sparsity ($s > 0.75$), but at lower sparsity the difference can be made up by adding more binary masks (which are cheaper to store). Some regularization benefits are lost.

We have also briefly explored a *hybrid* approach to understand the value of our linear combinations: inspired by early fine-tuning work where only the last layer of the network is retrained for each new task, we have created a hybrid setting where we employ our *ImpressLearn* algorithm in all layers except the last, and train a new SupSup mask in the last layer. This was done to demonstrate that the power of our linear combination routine does not come from the number of extra $\alpha$-parameters we allow for each new task. We show that allowing even more parameters in the last layer (for an additional SupSup mask) does not improve performance much, and that hence the power of our approach does not reside simply in parameter finetuning for new tasks (see Sec. A.5).

*Limitations:* Our experimental results show that *ImpressLearn* works well on several benchmarks, and particularly shines when the number of new tasks is large. In scenarios where the number of different tasks is *small* (like SplitCIFAR with 20 tasks only) our approach will only give limited parameter savings, if any. In the case of SplitImageNet we were only able to evaluate our approach for a relatively small number of masks and could not match benchmark performance. However, in particular in this case, our parameter savings are particularly impressive and highlight the power of transfer learning in this setting.

In future work, we plan to explore other settings where the linear combination optimization could be applied to leverage transfer learning for catastrophic forgetting.

## REPRODUCIBILITY

In keeping with conference guidelines, we have tried to ensure that our experimental results were free of any bias that would misconstrue the results. As specified in the paper, all experiments were run over multiple random seeds, on a variety of benchmarks using publicly available software and hardware. The code for the experiments is self contained and references appropriate instructions to install the relevant software dependencies. A copy of the experimental code will be made publicly available.

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

# A   APPENDIX

## A.1   HYPERPARAMETERS AND DESCRIPTION OF EXPERIMENTS

Here we provide the details of our experimental work.

Table 3: Dataset, Model and Hyperparameter Overiew.

| Model | Dataset | $|\mathcal{M}_{hom}|$ | LR | ($\lambda$) | Opt. | Batch Size |
|---|---|---|---|---|---|---|
| LeNet 300/100 | MNIST | | | | | |
| | Rotated | 250 | 0.002 | 0 | RMSprop | 128 |
| | Permuted | 250 | 0.002 | 0 | RMSprop | 128 |
| ResNet-18 | Split CIFAR 100 | 75 | 0.02 | 0.005 | ADAM | 64 |
| ResNet-50 | SplitImageNet | 75 | 0.0025 | 0.005 | ADAM | 96 |

For *homogeneous* basis masks, $|\mathcal{M}_{hom}|$ is the largest mask set we tried. For ADAM, momentum was set to $0.9$ and weight decay to $0.1$.

We performed experiments for a range of densities $s \in \{0.05, 0.08, 0.1, 0.15, 0.2, 0.25, 0.3\}$. For consistency, we maintained the same numerical ordering of tasks from each dataset across all experiments. Each seed varied the random initialization of the popup scores and the training data ordering. For a heterogeneous basis set we used one seed to generate the entire basis set. In the homogeneous case we seeded every mask to ensure mask diversity. The backbone network was fixed for one basis set and $\alpha$-optimization for new tasks. All runs were performed on three different backbone networks and train/test splits to ensure that results were sufficiently general. The boxplots contain averages over all these settings and sparsities.

### A.1.1   INFRASTRUCTURE

For all models except LeNet 300-100, we used GPU enabled hardware to expedite training time. Our experiments were performed on a SLURM cluster enabled with NVIDIA V100 Tesla and RTX 8000 GPUs. For the Imagenet experiments specifically, we were memory constrained and thus hard to tune batch size appropriately to fit within the 48 GB available on the device.

## A.2 IMAGENET RESULTS

Here we show *ImpressLearn* on SplitImageNet.

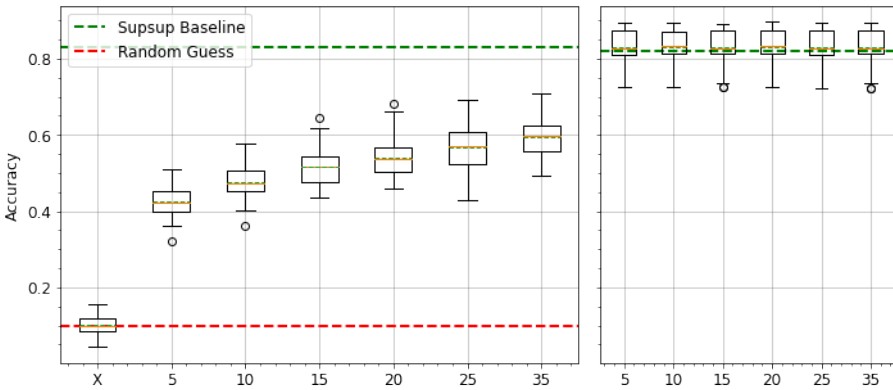

Figure 5: Left: Average performance on 5 new SplitImageNet tasks by number of *heterogeneous* impressions. $X$ indicates incorrect mask. Right: Performance of our $\alpha$-optimization in the *basis*-**GN** regime averaged over $min(|\mathcal{M}|, 25)$ basis tasks.

## A.3 BASIS GN SETTING AND A NEW TASK-INFERENCE ALGORITHM

As shown in the experimental results in Figs. 2, 4, 3 and 5, our $\alpha$-optimization is able to perform well when presented with a *basis*-task without task label. In that case, our algorithm finds a linear combination of masks that nearly always matches or outperforms the corresponding basis mask for this task. Our algorithm hence constitutes an alternative to entropy-based methods presented in (Wortsman et al., 2020) for the **GN** scenario (task labels given at training but not at inference).

In Fig. 6 we show how our algorithm outperforms the baseline *SupSup* performance for `PermutedMNIST`.

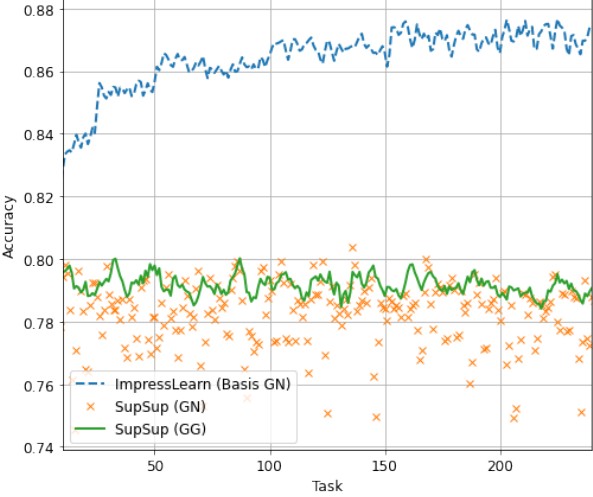

Figure 6: Comparison of inference accuracy between *basis*-**GN**, *SupSup* **GG** and *SupSup* **GN** on `PermutedMNIST`. **GN** performance averaged over 10 different data splits and orderings per seed and sparsity. All results are averaged over 3 different seeds and masks of density $\in \{5, 8, 10, 15, 20, 25, 30\} \times 10^{-2}$.

## A.4 RANDOM BASIS MASKS

Here we compare performance of a set of *random* basis masks to homogeneous masks to illustrate the trade-offs in case the SupSup step is not available or difficult to implement (Figs. 7, 8 and 9). Generally, in the limit of a large number of masks, the random methods, while yielding inferior accuracy, is faring surprisingly well and could become an alternative for resource-constrained settings.

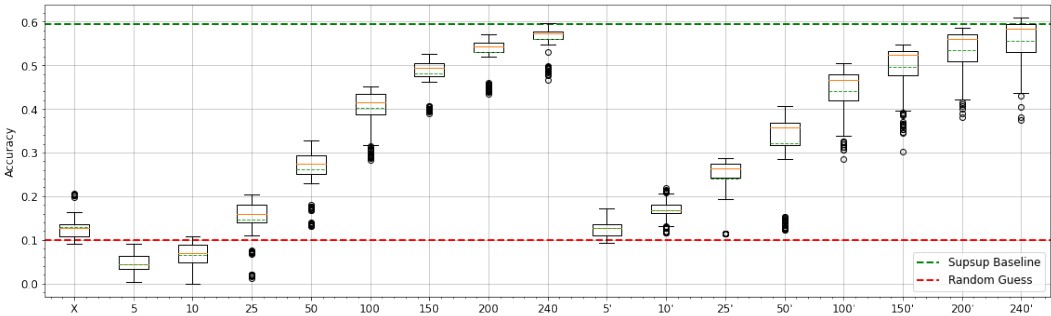

Figure 7: **Random vs Homogeneous masks:** Average validation performance on 10 unseen Rotated MNIST tasks by number of impressions and impression type. $X$ indicates incorrect mask. Standard numbers indicate an impression set of *random, untrained masks* and $N^{\wedge}$ indicates a homogeneous impression set. All results averaged over 3 different seeds and masks of density $\in \{5, 8, 10, 15, 20, 25, 30\} \times 10^{-2}$.

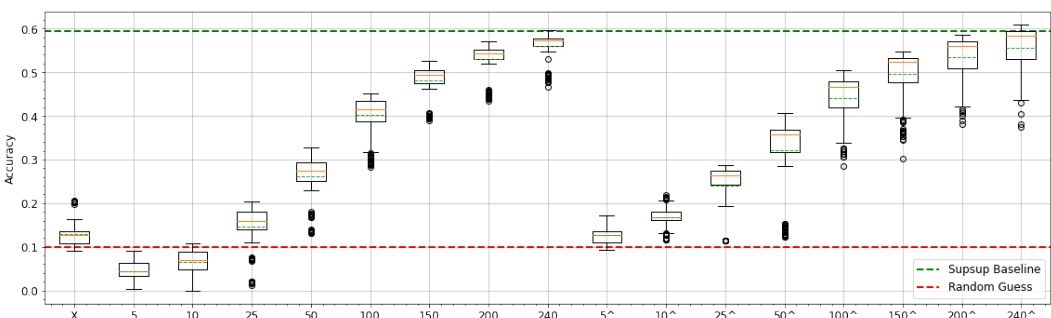

Figure 8: **Random vs Homogeneous masks:** Average validation performance on 10 unseen Permuted MNIST tasks by number of impressions and impression type. $X$ indicates incorrect mask. Standard numbers indicate an impression set of *random, untrained masks* and $N^{\wedge}$ indicates a homogeneous impression set. All results averaged over 3 different seeds and masks of density $\in \{5, 8, 10, 15, 20, 25, 30\} \times 10^{-2}$.

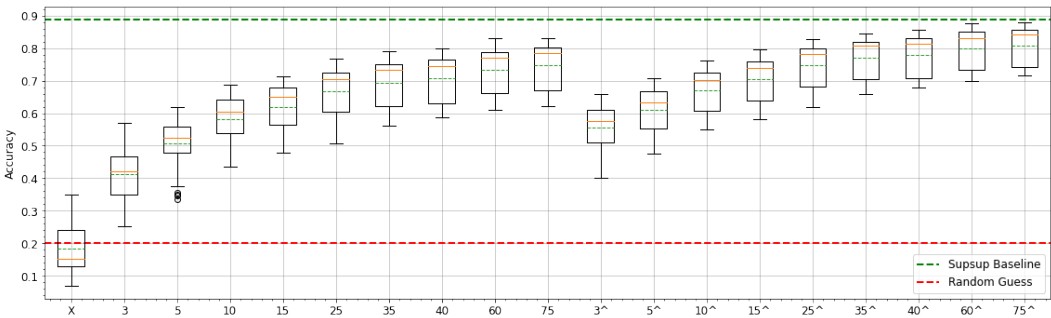

Figure 9: **Random vs Homogeneous masks:** Average performance on 5 unseen Split CIFAR-100 tasks by number of impressions and impression type. $X$ indicates incorrect mask. Standard numbers indicate an impression set of *random, untrained masks* and $N^{\wedge}$ indicates a homogeneous impression set. All results averaged over 3 different seeds and masks of density $\in \{5, 8, 10, 15, 20, 25, 30\} \times 10^{-2}$.

## A.5 HYBRID APPROACH

Here we study the hybrid setting where for each new task we employ our *ImpressLearn* algorithm in all layers except the last, and train a new SupSup mask only on the last layer. This demonstrates that the power of our linear combination routine does not come from the number of extra $\alpha$-parameters we allow for each new task. We show that allowing even more parameters in the last layer (for an additional SupSup mask) does not improve performance much, and that hence the power of our approach does not reside simply in parameter finetuning for new tasks (Fig. 10)

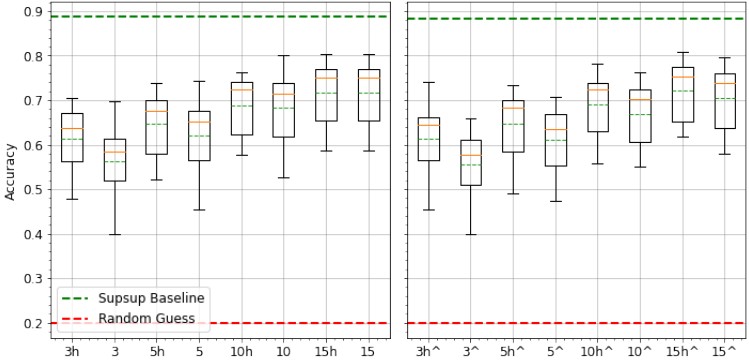

Figure 10: Left: Hybrid vs Regular (Heterogeneous Impressions). Right: Hybrid vs Regular (Homogeneous Impressions). Average performance on 5 new Split CIFAR-100 tasks by number of impressions and impression type. $h$ denotes a hybrid model was used whereas $^{\wedge}$ indicates a homogeneous impression set. $h^{\wedge}$ indicates both. All results averaged over 3 different seeds and masks of density $\in \{5, 8, 10, 15, 20, 25, 30\} \times 10^{-2}$.

