# OpenReview forum: "ImpressLearn: Continual Learning via Combined Task Impressions"
_ICLR.cc/2022/Conference — ICLR 2022 Submitted_

### Official Review · Reviewer_Xchu · 2021-10-28

**Correctness:** 3
**Technical Novelty And Significance:** 3
**Empirical Novelty And Significance:** 2
**Recommendation:** 5
**Confidence:** 3

**Details Of Ethics Concerns:**

The proposed approach shares the common concerns of ML systems, but there is no specific extra concern.

**Main Review:**

## Strength

The paper clearly describes the main technical proposal of applying linear combination of masks to randomly initialized networks. The approach seems to efficiently scale to a larger number of tasks in the given problem setup. Together with the new homogeneous impression approach, the paper seems to describe unique and novel technical ideas to improve on SupSup (Wortzman 2020).

## Weakness

Although the approach seems to novel, there are several weaknesses in the current paper.

The first is the performance degradation compared to SupSup baseline in CIFAR / ImageNet benchmarks. Although Sec 5 discusses the limitation when there are small number of tasks, Fig 4 seems to indicate a serious performance disadvantage compared to SupSup, which does not look reasonable. Considering that MNIST variants are artificial benchmarks and not practical in any application, the performance degradation in CIFAR / ImageNet limits the significance of the work.

Another missing comparison is the Hopfield network in SupSup, as discussed in Sec 2.

I wonder how strong is the catastrophic forgetting if we apply one of the discussed approaches in Sec 2; i.e., regularization-based, replay, or parameter isolation methods. As this paper only concerns the comparison to SupSup (fairly incremental), I feel it is unclear how impactful this work is in the broader continual learning context.

Regarding the task setup, which also applies to Wortzman 2020, I feel impractical to only consider the fixed l-way classification from the same or similar domain. At least the setup should consider a variable number of categories per task.

## Other

- What is the solid / dashed lines within the boxes in Fig 2, 3, 4, and 5? Median and mean?

**Summary Of The Paper:**

The paper describes an approach to continual learning by randomly initialized deep networks. The main idea is to extend the SupSup (Wortzman 2020) and apply linear combination of binary masks for efficiency. The paper also considers learning from multiple instances of the same task. The evaluation results indicate reasonable performance for the reduced parameters in common benchmarks.

**Summary Of The Review:**

While the proposed idea seems novel and the results seem to indicate good efficiency in terms of model footprint, the results show performance degradation to the baseline, and I feel the submission has rooms for improvement for acceptance due to the narrow focus on comparison to SupSup, as pointed in the main review. In overall, I do not think the paper reaches the acceptance threshold.

---

> ### Author Response · Authors · 2021-11-22
> **Authors' Response**
>
> We thank the the reviewer for their feedback. It is unfortunate that large-scale computer vision benchmark datasets such as Split-CIFAR-100 and Split-ImageNet, as they are commonly used in continual learning, support only a limited number of different tasks, where ImpressLearn is unlikely to shine by design. The practicality and performance of ImpressLearn come with an increasing number of tasks where more basis masks become available and where other “masking” approaches require significant memory resources.
>
> It is true that we do not compare ImpressLearn to other benchmarks from the continual learning literature and that adding these could improve the paper. Nevertheless, since our method is largely derived from SupSup and can be considered a resource-efficient extension of it, we thought that comparing ourselves to SupSup might be sufficient.
>
> Figures 2, 3, 4, and 5 display standard box plots; the green line represents the mean, while the orange line represents the median.

---

> > ### Comment · Reviewer_Xchu · 2021-11-30
> > **Final rating**
> >
> > Thanks for the clarification. While this work is incremental to SupSup, I think it is insufficient to benchmark with a single existing approach because that would fail to properly contextualize this work among literature. At least it is the authors' responsibility to justify in the main paper why SupSup can be considered the sole baseline and other approaches can be ignored, for the broader audience. Given the current quality, my final rating remains.

---

### Official Review · Reviewer_7LBU · 2021-11-01

**Correctness:** 3
**Technical Novelty And Significance:** 2
**Empirical Novelty And Significance:** 2
**Recommendation:** 3
**Confidence:** 4

**Details Of Ethics Concerns:**

The work does not involve ethical aspects.

**Main Review:**

Positive aspects:
- The proposed approach is scalable to a large number of tasks with a small parameter overhead than existing methods.
- The paper is in general clearly written
- The related work section covers most of the relevant papers in the field

Negative aspects:
- The scientific novelty of the paper is limited. It is an incremental work, relying heavily on [Wortsman et al., 2020]
- The experimental validation is limited and not convincing. The paper is not compared with any other state-of-the art methods.

**Summary Of The Paper:**

In the current paper, the authors propose a novel way to adapt an existing continual learning algorithm, leveraging principles from transfer learning.  More concretely, they learn an initial set of masks (or impressions) from a small number of basis tasks, and they use afterwards linear combinations of these masks in the learning process of new tasks. Therefore, their approach is able  to generalize to new tasks, allowing for scalable and parameter efficient continual learning (with much lower parameter overhead than existing methods).


**Summary Of The Review:**

Please find below some of my concerns:
1. First of all, please state clearly in the introduction in which aspects your paper is different from [Wortsman et al., 2020]
2. In page 3, you mention for the first time the term: GN-setting. Where is the meaning of 'GN'?
3. Please clarify the following statement (section 3): "It then generates the mask M by setting the top s fraction of scores in S to 1 and the rest to 0". How is this fraction chosen: heuristically, or do you use any criteria?
4. Regarding the experimental settings:
- State clearly, for each dataset, how do you define the tasks. How many basis tasks do you have for each dataset? How comes that for PermutedMNIST you have 250 learned tasks and the maximum could be 784 (Table 1)?
- What is the relationship between number of masks and number of basis tasks: you have one mask per one basis task?
5. Plots 2,3 and 4 are not clear
6. Figures 3 and 4: min(jMj; 25) basis tasks. How did you choose 25? Why do you use this criterion for basis tasks?
7. Please use a standard evaluation methodology in order to be able to interpret your results
8. Compare your approach against (at least) the following related methods: Packnet, Piggyback, [Wortsman et al., 2020], Ternary Feature Masks (TFM) [Masana et al., 2021]

M. Masana, T. Tuytelaars, and J. van de Weijer. Ternary Feature Masks: zero-forgetting for task-incremental learning. Proc. of CVPRW 2021, Workshop on Continual Learning

---

> ### Author Response · Authors · 2021-11-22
> **Authors' Response**
>
> We thank the reviewer for their comments and concerns, which we address below.
>
> 1. We discuss some key differences between ImpressLearn and SupSup in Section 1, and, in particular, in the paragraph titled “A basis of masks”.
>
> 2. GN is a term coined by Wortsman et al. (2020) that stands for Given/Not given to mean that task identifiers are available during training but not at inference. We explain the term on the same page.
>
> 3. It is up to a practitioner to choose $s$. In our experiments we try a few different values (reported in the captions of Figures 2, 3, 6, 7 and 8) and average the results over them.
>
> 4. The datasets used in our study are standard benchmarks in the continual learning literature. Table 1 states that the number of tasks in PermutedMNIST is, in fact, 784! (note the factorial sign) and not 784. In the heterogeneous scenario, we indeed generate one basis mask per seen task. We vary the number of basis tasks (masks) as seen in Figures 2, 3 and 4, for example.
>
> 5. Could you please specify what you found unclear?
>
> 6. We cap the number of evaluation tasks at 25 for convenience.

---

### Official Review · Reviewer_MxHs · 2021-11-01

**Correctness:** 4
**Technical Novelty And Significance:** 3
**Empirical Novelty And Significance:** 2
**Recommendation:** 5
**Confidence:** 3

**Main Review:**

The paper is in general well-written and structured.

The proposed method follows the popular approach of alleviating forgetting through masking neurons for different tasks. As mentioned by the authors this has been explored in prior work such as PackNet and Piggyback, SupSup. Related work like [1,2] are not discussed. The main contribution over prior work is that this paper introduces a subspace of masks, that is spanned by a mask basis, which is referred to as “impression”.

My concern is that the contribution w.r.t. prior works is rather slim. In the experimental evaluation, the paper only compares to a single baseline, SupSup, and ignores the results of all other standard baseline methods mentioned in the related work, including replay and regularization-based methods, which achieve significantly higher performance on the evaluated benchmarks (e.g. FROMP [3] achieves >90% on P-MNIST). The contribution w.r.t. Supsup remains unclear beyond a reduction of the number of parameters, which comes at the cost of a moderate loss in performance. I would recommend the authors to motivate more clearly why the reduction in parameters is a problem of critical importance.

[1] Sangwon Jung, Hongjoon Ahn, Sungmin Cha, and Taesup Moon. Continual learning with node-importance based adaptive group sparse regularization, 2020.
[2] Chungkuk Yoo, Bumsoo Kang, and Minsik Cho. SNOW: Subscribing to Knowledge via Channel Pooling for Transfer & Lifelong Learning of Convolutional Neural Networks. 2019
[3] Pan, P., Swaroop, S., Immer, A., Eschenhagen, R., Turner, R. E., and Khan, M. E. Continual deep learning by functional regularisation of memorable past, 2021.


**Summary Of The Paper:**

The paper studies continual learning and that context the important and prominent problem of catastrophic forgetting. Similar to related work the paper achieves this through learning task-dependent masks on the neural network representation (effectively generating different sub-networks for different tasks).  The main contribution w.r.t. related work is that the masks are constrained to a sub-space which is spanned by a "basis of masks". The experiments show some reduction in the number of required parameters over a single baseline method.

**Summary Of The Review:**

Overall, the contribution over related work seems limited to me, but I am open to feedback from the authors if my initial perception is not correct.

---

> ### Author Response · Authors · 2021-11-22
> **Authors' Response**
>
> Thanks Reviewer MxHs for the time spent reading out paper and your comments. The “masking” approaches to continual learning, such as SupSup and Piggyback, require allocation of parameter masks for each additional task, which might become impractical in scenarios where the ultimate number of tasks is large. ImpressLearn, on the other hand, reduces this burden by keeping the number of masks fixed and requiring truly minimal additional per-task costs. Hence, the main advantage of ImressLearn is resource efficiency in these scenarios, i.e., where the number of tasks grows without bound. We mention this motivation in the abstract and in Section 1 (the paragraph titled “a basis of masks”).

---

### Official Review · Reviewer_Tj8S · 2021-11-08

**Correctness:** 3
**Technical Novelty And Significance:** 2
**Empirical Novelty And Significance:** 2
**Recommendation:** 3
**Confidence:** 5

**Main Review:**

Comment:

1: The proposed approach is parameter efficient and requires very few parameters for each novel task (assuming that basis masks are learned)

2: The paper follows the idea of SupSup, and the difference is how to combine the previously learned mask for the novel task. Also, the task prediction idea is the same as SupSup. Another contribution is learning homogeneous masks, which may be helpful in many scenarios, but the basic idea is the same. Overall, it seems that the paper has limited contribution compared to SupSup.


3: the models are evaluated over the smaller architecture ( i.e. LeNet), and it's challenging to evaluate the proposed model using only the given experiment. For the larger architecture, only parameter efficiency are reported (Table-2). I would suggest the author, please report the result for the ResNet-18 architecture over the CIFAR-100 and Mini-ImageNet datasets.

4: The results in Fig-3 are confusing. In the SupSup paper (Fig-3 SupSup paper), the result for the PermutedMNIST is ~94% (GN Scenario), while the paper reported the SupSup result of ~80%. How author got this result? As compared to the SupSup, the performance of the proposed model seems much lower. A similar pattern we observe for the CIFAR100 dataset, in Fig-4 performance is much poorer and for the GN scenario model only performs well for the very small task sequence (5, from Fig-4 right).

5: What architecture is used for the ImageNet dataset experiment?

6: The performance of the proposed model highly depends on the number of learned base tasks. Sometimes we have only a 5/10 task sequence, and if for the small task sequence, we learn 50 or 100 task sequence, it is much costly.  Also, we can observe that the homogeneous setting model does not perform well. Therefore, for the limited task sequence model may not be helpful.

7: The results are only compared with the SupSup paper. I would suggest to the author; please consider the recent expansion based model [a,b,c] for the comparison.

[a] Efficient Feature Transformations for Discriminative and Generative Continual Learning, CVPR-21

[b] Ternary Feature Masks: zero-forgetting for task-incremental learning, CVPRW-21

[c] Continual Learning using a Bayesian Nonparametric Dictionary of Weight Factors, AISTATS-21


**Summary Of The Paper:**

The paper learns the binary basis mask for a few task sequence tasks, later linear combinations of the basis mask can be used as a mask for the new task. The parameter learned for combining the basis mask is significantly fewer than learning a novel mask for each new task sequence. The paper also proposes a homogeneous mask learning. For most of the datasets, we don't have the flexibility to learn an enormous basis mask and model performance highly depends on the number of learned basis tasks. The proposed model is evaluated over the MNIST, CIFAR100 and Split-ImageNet dataset over the smaller architecture.

**Summary Of The Review:**

The paper is lagging in experimental results with the recent baseline. Also, the contribution is limited compared to SupSup.

---

> ### Author Response · Authors · 2021-11-22
> **Authors' Response**
>
> We thank the reviewer for their feedback and would like to address their questions and concerns.
>
> 3. We do report experimental work on larger datasets (Table 3 and Figures 5, 9, and 10).
> 4. Results reported in Figure 3 are averaged across masks of different sparsity. While Wortsman et al. (2020) do not disclose the sparsity level used in Figure 3 of their paper, the discrepancy in our results can potentially come from working in different sparsity regimes.
> 5. We use ResNet-50 as stated in Table 3 (Appendix A.1)
> 6. It is true that ImressLearn is more practical in scenarios with many tasks and this setting is our main motivation. However, homogeneous masks can sometimes generate enough “impressions” from a single task without much loss in performance.
> 7. We regard our contribution to be a resource-efficient extension of the idea introduced by Wortsman et al. (2020) and, hence, it is our main baseline. However, we agree that a more complete comparison should be drawn in future research.

---

> > ### Comment · Reviewer_Tj8S · 2021-11-30
> > **FInal Comment**
> >
> > Thanks for your response. The author partially addresses my concern.
> > The result reported on the larger datasets are appreciated, but still, I believe that the paper is not ready to publish.
> > I will keep my initial rating, also suggest to the author please compare the proposed model with a few recent models that follow the same setting.

---

### Decision · Program_Chairs · 2022-01-20

**Decision:**

Reject

**Comment:**

This paper presents an approach "ImpressLearn" to continual learning using the idea of task-specific masks. The idea builds upon another idea - SupSup (Wortzman 2020) - which uses a backbone network shared by all the tasks and binary task-specific masks. However, the number of parameters for an approach like SupSup can become excessively large when the number of tasks is very large. This paper presents a solution by having a small number of basis-masks and learning a weighted combination of these basis-masks to use as the task-specific mask for each task. The experimental results show that ImpressLearn yields significant parameter savings as compared to SupSup.

There were several concerns shared by all the reviewers, such as (1) Limited novelty as compared to SupSup, and (2) Limited experimental evaluation and not having enough baselines. From my own reading of the paper, I largely agree with the assessment of the other reviewers.

The authors responded to the original reviews and acknowledged some of the concerns raised by the reviewers. The reviewers read the authors' response but their assessment has remained unchanged.

The basic motivation and the idea is nice but offers limited novelty (especially as compared to SupSup). If the authors could improve the experimental evaluation (more baselines, larger datasets/networks, etc), it will be a much stronger paper. However, in its current shape, I as well as the other reviewers do not think that the paper is ready for publication.